# Comparative Investigation of Aquatic Invertebrates in Springs in Münsterland Area (Western Germany)

**Sura Abdulghani Alqaragholi [1,\*], Wael Kanoua [2,3] and Patricia Göbel [1]**

[1]   Institute of Geology and Palaeontology, University of Münster, 48149 Münster, Germany;
      pgoebel@uni-muenster.de

[2]   Department of Hydrogeology, TU Bergakademie Freiberg, Gustav–Zeuner Str. 12, 09599 Freiberg, Germany;
      wael_kanoua@yahoo.com

[3]   Department of Petroleum Engineering, Chemical and Petroleum Engineering Faculty, AL Baath University,
      Homs, Syria

\*   Correspondence: s_alqa01@uni-muenster.de

**Abstract:** The main aim of this study was to investigate the abundance of invertebrates in groundwater in relation to groundwater conditions (groundwater table, discharge, rainfall, and physio-chemical parameters), and to examine the suitable time for invertebrate sampling in springs. Thus, eight springs in two separate study areas, "Baumberge" and "Schöppinger Berg" (Münsterland area, North-Rhine Westphalia in Germany), were sampled five times (24 h for 2–5 consecutive sampling days) between November 2018 and October 2019. The results showed high spatial and temporal variance. In general, the existence of invertebrates and stygobites increased, whereas invertebrate types decreased with increasing hydraulic head and spring discharge. Therefore, investigating the abundance of invertebrates and invertebrate species is recommended to be done separately. Abundance of invertebrates was affected by different factors in both areas. Spearman correlation test (two-tailed) and factor analyses ($n = 80$, $p \leq 0.01$) highlighted the importance of detritus as the main controlling factor for invertebrate existence and stygobite individuals in Baumberge, whereas dissolved oxygen is essential for their existence in Schöppinger Berg.

**Keywords:** invertebrates; abiotic variables; sampling time; springs; Germany



## 1. Introduction

Groundwater is the most important natural resource, particularly in areas where surface water is scarce [1,2]. It is of vital importance for daily life, food production, and manufacturing [3,4]. Thus, studying the chemical and physical parameters of groundwater [5,6], the physical characteristics of the hosting aquifers [7], the relationships and interaction between groundwater and the surrounding rock material [8], the interaction between surface- and groundwater [9–11], and managing this valuable source quantitatively and qualitatively [4,12] are important topics that have attracted the attention of numerous scientists in recent decades.

The groundwater ecosystem is characterized by total darkness and often low organic carbon [13,14], thus it usually has relatively stable conditions and a physically inactive environment [15], and undergoes spatial and temporal changes [14,15]. Various kinds of aquifers are reservoirs of groundwater and also host different groundwater invertebrate (stygofauna) communities [16–20]. These stygofauna are considered a biomonitor of groundwater quality and can be used to track and monitor sources of pollution [21–24]. The invertebrate species can also be considered descriptors of the aquifer type, habitat structure, water flow regime, and groundwater flow paths [25].

Because human activities significantly disturb the natural state of the environment, efforts to develop different monitoring methods started almost a century ago [26]. Subsequently, invertebrates were utilized to assess environmental conditions in aquatic ecosystems [27]. In this regard, understanding environmental factors that influence groundwater

dwelling invertebrates is crucial for groundwater ecosystem monitoring [28]. This biological monitoring by means of aquatic invertebrates depends on the fact that different invertebrate taxa respond differently to different extents of pollution, and their respective responses can be used to study water quality.

Faunal distribution in groundwater is an important issue, and distribution patterns are directly linked to spatial and temporal heterogeneity of groundwater and the hosting aquifer [29]. This heterogeneity is affected by many factors, e.g., groundwater recharge [14], river flow [16], cropping [30], and irrigation cycles [31], and what these factors deliver into the groundwater, such as dissolved oxygen (DO), organic matter [32], and nutrients [33]. These additives to groundwater positively and negatively affect the life of different fauna communities [34]; in other words, they either enhance life by catering sustainably for ubiquists, or render the life of uncompetitive communities difficult or impossible. Thus, this impact needs to be taken into account when sampling groundwater fauna communities and when evaluating monitoring studies.

Therefore, efforts have been directed to describe the distribution of invertebrates in groundwater systems, and to reveal the relationship between this distribution and different environmental parameters. These efforts help in the formulation of hypotheses about the behavior of groundwater fauna to any variation (e.g., hydraulic head, hydrological regime, land use and land cover, anthropogenic effect, and water quality) in the hydrogeological system under consideration [15,28,35–43]. In general, stygofauna were found to exist under diverse physicochemical conditions of groundwater systems, e.g., electrical conductivity 11.5–54,800 μS/cm; temperature 17.0–30.7 °C; and pH 3.5–10.3 [44].

Natural and anthropogenic gradients in groundwater quality influence stygofauna assemblages [21]. Consequently, one of the most important questions to be answered is the extent to which faunal communities respond to environmental gradients, and how their existence is linked to other groundwater parameters throughout the year. Seasonality in this regard determines the suitability of a habitat for a taxon, and biotic interactions determine the success of a taxon in a particular location [42]. Taking all of these factors into consideration, it is important from economical and time-saving perspectives to effectively select the best time to undertake sampling, which appears to be site-specific.

Therefore, the current study investigates and monitors invertebrates, particularly stygobite biota, in small natural perennial springs as the outlets of a fractured aquifer in the connected catchment area. This research examines the seasonality signature (existence, abundance, and variability) of invertebrates in groundwater in two study areas, in addition to the variance of some groundwater on-site parameters during one sampling year. One key outcome that was determined is the best time to carry out invertebrate sampling in springs.

## 2. Materials and Methods

### 2.1. Study Area

The two study areas are located in Münsterland region in North-Rhine Westphalia (NW Germany). The areas "Baumberge" (BB) and "Schöppinger Berg" (SB) are located 15 km from each other. Both areas were influenced by the last glacial maximum [45]. The area is fractured and slightly karstified and consists of porous marlstone forms. Due to the relief inversion, a bowl-like structure was formed, which can therefore be considered as a closed and isolated, yet hydrogeologically uniform, groundwater system. Both areas (BB and SB) are affected by surface water only from rainwater because surface water is absent in the areas.

#### 2.1.1. Groundwater System Baumberge

Baumberge (BB) is an agriculturally dominated and forested mountain ridge with an area of 40 km$^2$, a maximum height of 186 m above mean sea level (m a.s.l.), and an extension of 15 km NW–SE and 4 km SW–NE. It rises 100 m above the surrounding flat and forms a precipitation barrier.

The special geological structure was already described by [46]. In BB, the upper Baumberge Formation consists of sand, marl, and lime marlstone with medium permeability; the lower Coesfeld Formation consists of clay and lime marlstone with very low water permeability. At the geological boundary between Baumberge and Coesfeld Formations, up to 25 perennial natural single springs and ca. 20 spring areas (with up to 100 intermittent natural outlets) are formed as overflow springs on all sides of the BB along the horizon between 100 and 120 m a.s.l. In the center of the BB structure the minimal depth of the groundwater table is 60 m, with an average annual groundwater fluctuation of 10 m, depending on the groundwater recharge. Groundwater is of the Ca-HCO$_3$ type, with anthropogenic effects reflected by inputs of nitrate (NO$_3$) and sulphate (SO$_4$).

### 2.1.2. Groundwater System Schöppinger Berg

Schöppinger Berg (SB) is an agriculturally-dominated mountain ridge with an area of 18 km$^2$, maximum height of 157.6 m a.s.l., and an extension of 7 km NW–SE and 3 km SW–NE. Its summit reaches up to 157.6 m a.s.l., which is, therefore, only 80 m above the surrounding areas [47]. It branches from the BB mountains to the north and has a similar geological structure. In SB, the upper Coesfeld Formation consists of clay and lime marlstone with low water permeability; the lower Holtwick Formation consists of clay and silty marlstone with very low water permeability. At the boundary between Coesfeld and Holtwick Formations, up to eight perennial spring areas (with up to 60 almost natural outlets) and two intermittent single springs are formed as overflow springs on all sides of the SB along the horizon between 78 and 87 m a.s.l. In the center of the SB structure the minimal depth of the groundwater table is 47 m, with average annual groundwater fluctuation of 6 m. Groundwater in this area is of the Ca-HCO$_3$ type, with anthropogenic input reflected by a high concentration of SO$_4$, and NO$_3$ concentrations of about 50–70 mg/L.

### 2.2. Methodology

Eight groundwater springs were sampled (Supplementary Table S1). All springs are natural springs and groundwater flows out all year round (perennial) at selected and clearly recognizable, locally limited outlets, and immediately forms visible runoff (rheocrene). The Longinus Tower well (latitude (lat.): 51°57′36″ N, longitude (lon.): 7°21′56″ E) was selected as a representative groundwater observation well in the center of BB. The springs are distributed in two groups along BB and SB regions (Figure 1).

### 2.2.1. Field Work

The sampling survey and spring monitoring were carried out from November 2018 to October 2019, with a sampling frequency of two to three months. The sampling campaigns were carried out in November 2018, January 2019, April 2019, July 2019, and October 2019. In each sampling campaign, the sampling was conducted at each spring for several consequent days ($n$d: two to five days sampling period). Overall, there were 160 spring sampling events. This sampling of several days was planned, first, to avoid short-term fluctuations and, second, to overcome the problem with the sampling technique "fauna net" and the accessibility to the sampling sites.

Spring water with invertebrates, sediment, and detritus was captured with plankton nets, with a mesh size of 74 μm, modified after [48]. The plankton nets were installed against the selected outlet at a representative discharging point of each spring. After 24 h, the captured samples were washed and filtered on site with a mesh size of 74 μm. This mesh size is suitable for collecting invertebrates of a wide range of body sizes, including meiofauna [49]. Gammaridae from surface water were removed from the filter mesh to prevent predation of invertebrates. After filtration, the samples were stored in 100–1000 mL flasks in a cool box under dark and cold conditions, and transported within half a day to the laboratory for the additional analysis (laboratory work) to be carried out. Directly after invertebrate sampling, physicochemical parameters and discharge were estimated

at the same discharging point of the spring. Physicochemical parameters of the spring water (pH value (pH; -), temperature (Temp.; °C), electrical conductivity (EC; μS/cm), and dissolved oxygen (DO; mg/L) were measured on-site using a multi-instrument (WTW 340i). The discharge of each sampled spring $V$ (L/s) was estimated by determining the liters of discharge that could be collected per second, with the help of a one-liter bucket and a stopwatch. The fraction of the estimated spring discharge that actually flowed through the net was also estimated. In each sampling campaign, the groundwater level ($h_{GW}$: m a.s.l.) was measured with a water level meter type 100 LTC (HT Hydrotechnik GmbH, Germany). Table 1 summarizes the preconditions of the field work undertaken in this study.

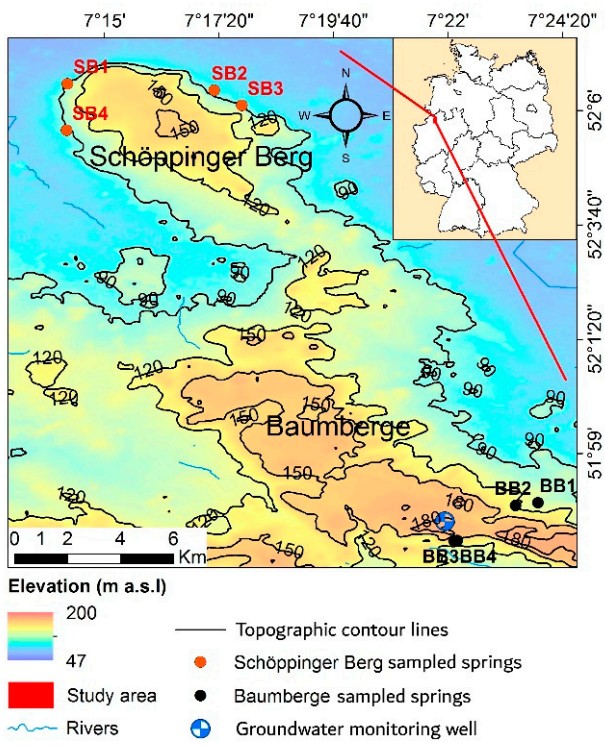

**Figure 1.** Upper right inset map shows Germany with border lines of its different states, and the study area as red rectangle. The main map shows the map of the study area with the location of eight sampling points/springs (black and red dots) in Baumberge and Schöppinger Berg, observation well (blue dot), main rivers and streams (blue lines), and contour lines, drawn on Shuttle Radar Topography Mission SRTM (PROCESSED SRTM DATA VERSION 4.1) digital elevation model (downloaded from CIAT-CSI SRTM website, on 20 August 2020).

**Table 1.** Precondition of field work for sampling eight springs in Baumberge and Schöppinger Berg between November 2018 and October 2019.

| Study Area | Sampling Campaign Data (Unit) | November 2018 | January 2019 | April 2019 | July 2019 | October 2019 |
|---|---|---|---|---|---|---|
| **Baumberge** | Sampling date | 07–09 | 14–17 | 23–26 | 01–05 | 15–18 |
| | Sampling period = number of days $nd$ (-) | 3 | 4 | 4 | 5 | 4 |
| | Groundwater table at the observation well $h_{GW}$ (m a.s.l.) | 114.3 | 114.9 | 122.8 | 116.3 | 114.24 |
| **Schöppinger Berg** | Sampling date | 08–09 | 14–17 | 23–26 | 01–05 | 15–18 |
| | Sampling period = number of days $nd$ (-) | 2 | 4 | 4 | 5 | 4 |
| | Groundwater table at the observation well $h_{GW}$ (m a.s.l.) | 114.3 | 114.9 | 122.8 | 116.3 | 114.24 |

### 2.2.2. Laboratory Work

Water samples collected from the springs contained a mixture of water, sediments of different grain sizes, detritus (organic matter), ochre, and invertebrates. These samples were stored at 10 °C in a dark refrigerator and processed in the laboratory within one week of collection.

Invertebrates and detritus were separated from water by passing the water sample through a planktonic net (mesh size 74 μm). Animal individuals were sorted alive to facilitate the observation and examination of the animals placed in petri dishes (5 and 7 cm) under a dissecting microscope with the help of disposable pipette and forceps. Firstly, animal individuals were sorted into stygobites (sb) fauna (smaller transparent individuals without eyes) and non-stygobites (non-sb) fauna (stygoxenes and stygophiles, bigger pigmented individuals with clearly identifiable eyes). Secondly, the fauna was determined at the following taxa groups: amphipods, cyclopoids, harpacticoids, ostracods, isopods, syncarida, nematodes, oligochaetes, diptera, turbellarias, gastropods, acari, and other insects according to DWA [50]. Thirdly, all individuals were stored separately in groups in small glass flasks of 10 mL, filled with 97% ethanol.

Detritus in the residual spring water sample was estimated with reference to the four levels according to [48]: absent (level 0; without sediment), little (level 1; small amount of sediment), much (level 2; up to 1 cm sediment in the flask), and very much (level 3; more than 1 cm sediment in the flask).

### 2.2.3. Biotic Data (Counting and Estimating of Invertebrates)

The existence of invertebrates (stygofauna) and stygobite individuals was described by the average quantity of invertebrates/stygobite individuals per day. The abundance of invertebrates/stygobite individuals was described by the average quantity of invertebrates/stygobites individuals per cubic meter.

The calculation of the average quantity of stygofauna individuals per day $\overline{I_{sf\_d}}$ (Ind./d) or stygobite individuals per day $\overline{I_{sb\_d}}$ (Ind./d) in one sampling campaign was undertaken by summing the quantity of stygofauna individuals found and counted per day $I_{sf\_d}$ (Equation (1)), and relating it to the number of sampling days ($n$d) as follows:

$$\overline{I_{sf\_d}} = \frac{\sum_{i=0}^{n} I_{sf\_di}}{n\mathrm{d}} \quad \text{or} \quad \overline{I_{sb\_d}} = \frac{\sum_{i=0}^{n} I_{sb\_di}}{n\mathrm{d}} \tag{1}$$

The calculation of the average quantity of stygofauna individuals per cubic meters $\overline{I_{sf\_V}}$ (Ind./m$^3$) or stygobite individuals per cubic meters $\overline{I_{sb\_V}}$ (Ind./m$^3$) in one sampling campaign was conducted by summing the quotients from the quantity of stygofauna individuals $I_{sf\_d}$ (Equation (2)) per day related to the estimated spring discharge flowing through the plankton net $V$ (L/s) extrapolated to one day. Finally, this sum was related to the number of sampling days ($n$d) as follows:

$$\overline{I_{sf\_V}} = \sum_{i=1}^{n} \frac{\sum I_{sf\_di}}{V \cdot 86,4} / n\mathrm{d} \quad \text{or} \quad \overline{I_{sb\_V}} = \sum_{i=1}^{n} \frac{\sum I_{sb\_di}}{V \cdot 86,4} / n\mathrm{d} \tag{2}$$

The ecological impact of surface water and rainwater on groundwater was estimated by calculating the groundwater fauna index (GFI) according to [48]. Using DO (mg/L), standard deviation (SD) of temperature $SD_{Temp}$ (°C), and the relative amount of detritus D (four level scale 0–3) estimated in the laboratory, the GFI could be calculated for each spring and sampling campaign as follows:

$$\mathrm{GFI} = \sqrt{\mathrm{DO}} \cdot \sqrt{\mathrm{D}} \cdot SD_{Temp} \tag{3}$$

According to [51], three classifications of stygofauna (Table 2) can be derived using the aforementioned formula.

**Table 2.** Values of GFI according to [51].

| Type | GFI | Invertebrates | Total Abundance [Ind./L] | Number of Taxa |
|------|-----|---------------|--------------------------|----------------|
| I | <2 | Often absent fauna; prevailingly stygobites individuals | <3 | <1 |
| II | 2–10 | Prevailingly stygobite individuals | <50 | 1–4 |
| III | >10 | Prevailingly ubiquists stygophile and stygoxene individuals | >50 | >3 |

2.2.4. Meteorological Data

Meteorological data, particularly rainfall, within the sampling period were adopted from [52] (Baumberge well: 11016300). The rainfall of each sampling campaign R (mm/30d) was calculated by the sum of daily rainfall within 30 days before the first day of each sampling period.

2.2.5. Data Processing

Data were missing (16%) due to numerous reasons. Handling of these missing data was performed by excluding them from the analyses. Prior to data analysis, a Mann–Whitney U Test was conducted, and it was found that the difference in physicochemical parameters between the two groups of springs was significant ($p \leq 0.01$), with the exception of pH; thus, the two groups of springs were taken separately and compared with each other. Statistical analyses of the data (mean, median, SD, and range with minimum and maximum) were conducted using R (R version 4.0.3, © The R Foundation), SPSS.26 (IBM Corp. Released 2019. IBM SPSS Statistics for Windows, Version 26.0. IBM Corp.: Armonk, NY, USA), and MS EXCEL 2016. It is worth mentioning that groundwater table was measured for one time in each sampling campaign. Thus, for the statistical analyses, groundwater table was considered constant for the sampling period of several consequent days within each sampling campaign. Spearman correlation ($n = 80$, $p \leq 0.01$) was conducted between different biotic and abiotic data collected during the sampling campaigns. Factor analysis (FA) was applied to reduce the number of variables [53]. A principal component analysis (PCA) was performed to extract principal components. To enhance the interpretability, a varimax rotation was applied to obtain high loadings of variables on a single factor. The downward curve scree plot (showing the number of factors versus the eigenvalues) was used as a cut-off for the number of factors to be extracted. The same result (number of factors) was also obtained when applying Kaiser's lower bound eigenvalue of greater than one. The significance of the factor loadings was obtained from the $p$-value of the Pearson correlation coefficient. Before processing, data vectors were normalized to a mean value of zero and an SD of one. The presence or absence of any influence of a special parameter on the investigated variables could be inferred from similarities and variations in the studied variables.

**3. Results and Discussion**

*3.1. Abiotic Condition*

3.1.1. Groundwater Table, Rainfall, and Spring Discharge

Data of the groundwater table and rainfall are presented in Table 3 at eight springs during the sampling period between November and October The groundwater table fluctuated considerably during the sampling months, with the highest value of 122.82 m a.s.l. in April and the lowest value of 114.24 m a.s.l. in October Rainfall showed a high amount of 114.85 mm/30d in January and a low amount of 22.5 mm/30d in April It is clear that the fluctuation of the groundwater table follows that of rainfall with a time delay.

**Table 3.** Groundwater table, rainfall, and discharge at eight springs during the sampling period between November 2018 and October 2019.

| Region | Sampling Campaign Parameters | November 2018 | January 2019 | April 2019 | July 2019 | October 2019 |
|---|---|---|---|---|---|---|
| Baumberge and Schöppinger Berg | Groundwater table at the observation well $h_{GW}$ (m a.s.l.) | 114.30 | 114.9 | 122.8 | 116.3 | 114.24 |
| | Average daily discharge of all springs $V$ (m³/d) | 16.8 | 15.8 | 18.8 | 14 | 16.1 |
| | Sum of 30-day rainfall before first sampling day (mm/30d) | 40.2 | 114.5 | 22.5 | 32.5 | 112.2 |

The average daily spring discharge showed a fluctuation between the highest value of 18.8 m³/d measured in April and the lowest value of 14.0 m³/d measured in July (Table 3). This fluctuation, however, to some extent followed the fluctuation of the groundwater table. Of note are the comparable discharges of approx. 16.2 m³/d in the months November, January, and October, in which the groundwater table was also at a comparable level of approx. 114 m a.s.l.

Looking at the average daily spring discharge of one spring, it becomes clear that the annual fluctuations in the single spring are not high (Figure 2 left), and similar for BB and SB (Figure 2 right). SB1 shows the highest average value of 47.6 m³/d and SB3 the lowest average value of 0.9 m³/d. In the following order: SB1, BB4, BB2, BB3, BB1, SB2, SB4, and SB3, a decrease in the spring discharge was determined.

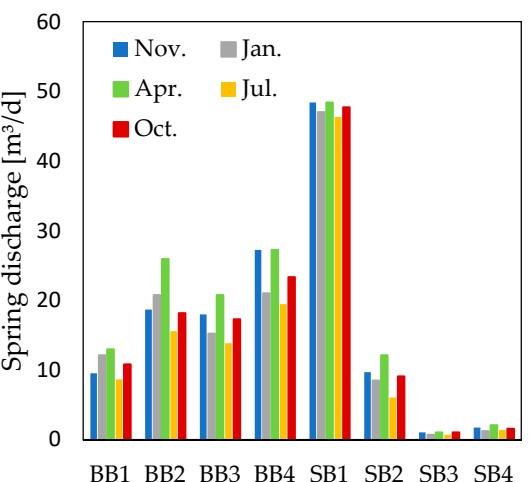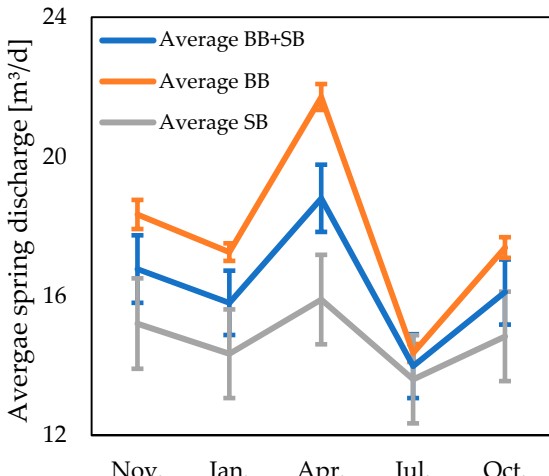

**Figure 2.** Average daily spring discharge (m³/d) for each sampled spring and sampling campaign (**left**) and for eight springs from November 2018 until October 2019 (**right**); vertical error lines represent the ±SD/10 of the respective parameter.

### 3.1.2. Spring Water Physicochemical Parameters

A statistical summary of the physicochemical parameters of spring water in the study area is presented in Table 4. All physicochemical parameters show spatio-temporal fluctuation, with higher amplitude in BB in the case of Temp., pH, and DO, and higher amplitude in SB in the case of EC and discharge. Compared to SB, the spring water in BB shows a slightly lower Temp. (with slightly higher fluctuation), similar pH values (with slightly higher fluctuation), lower DO (with similar fluctuation), lower EC (with lower fluctuation), and higher discharge (with lower fluctuations). Furthermore, it should be noted that the spring water was always clear and showed no turbidity, except in BB2 which showed heightened turbidity and phosphate values, a very high diversity of microbes, and a contamination with E. coli, which hints at an anthropogenic influence [54].

**Table 4.** Mean, median, SD, and range (minimum = Min., maximum = Max.) of physicochemical parameters measured at springs in Baumberge and Schöppinger Berg in five sampling campaigns from November 2018 to October 2019.

| Region | Statistics | Temp (°C) | pH | DO (mg/L) | EC (µS/cm) | Discharge $V$ (m³/d) |
|---|---|---|---|---|---|---|
| **Baumberge** | Mean | 10.0 | 7.2 | 6.0 | 729 | 25.3 |
| | Median | 9.9 | 7.2 | 5.9 | 727 | 24.2 |
| | SD | 0.5 | 0.2 | 0.8 | 15 | 6.3 |
| | Min. | 9.3 | 6.9 | 4.2 | 700 | 17.3 |
| | Max. | 12.4 | 8.1 | 8.6 | 758 | 43.2 |
| **Schöppinger Berg** | Mean | 10.5 | 7.2 | 6.7 | 778 | 20.5 |
| | Median | 10.1 | 7.2 | 7.1 | 771 | 8.6 |
| | SD | 0.7 | 0.2 | 1.2 | 22 | 22.9 |
| | Min. | 9.5 | 7.0 | 4.7 | 749 | 1.7 |
| | Max. | 12.0 | 7.8 | 8.7 | 825 | 60.5 |

The on-site parameters show clear differences between BB and SB (Figure 3). The value of EC was the highest in both areas in April and the lowest value was measured in July The value of pH in both spring groups was circumneutral pH = 7.2, which is normal for groundwater with no source of contamination (neither natural nor anthropogenic), with moderate fluctuation from month to month [55].

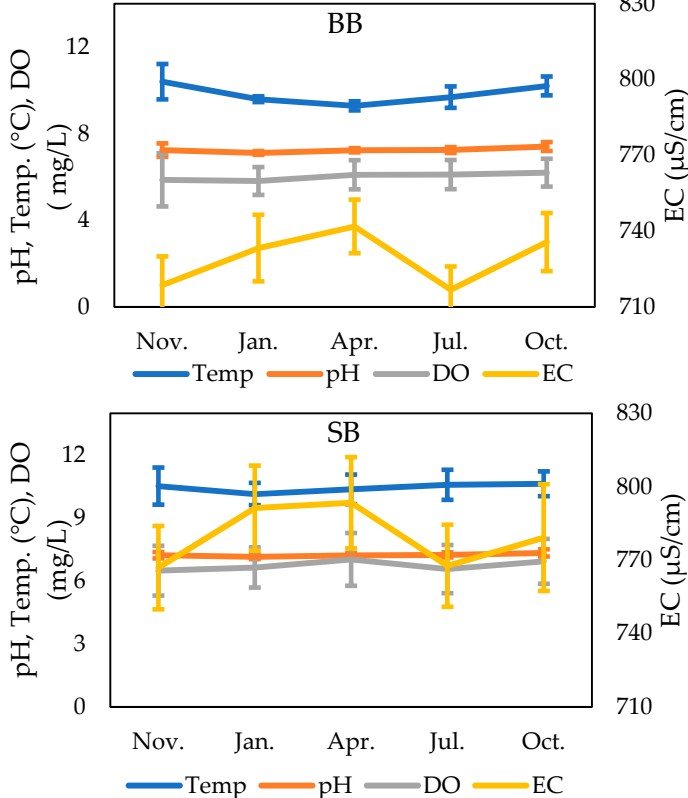

**Figure 3.** Temporal fluctuation of daily average physicochemical parameters of spring water (pH: brown line, Temp: blue line, DO: gray line, and EC: yellow line) in the study area from November 2018 until October 2019. Upper chart BB and lower chart SB; vertical error lines represent the ±SD of the respective parameter.

To estimate the potential existence of invertebrates, the GFI was calculated using abiotic parameters of each sampled spring and sampling campaign (Figure 4). The GFI was between 0.1 and 5.8, with the exception of SB4; most of the GFI values were below 2. Therefore, the groundwater system in the spring catchment areas is characterized by weak hydrological exchange with little food supply, and thus very low existence of individuals of the stygobite macrofauna [51]. In the order BB2, SB1, BB3, SB3, SB2 = BB1, and BB4, a decrease in invertebrates can be expected; only in SB4 can invertebrates not be expected. Among the temporal fluctuations, January appears to have the least potential for the existence of invertebrates, whereas July and October have the highest potential.

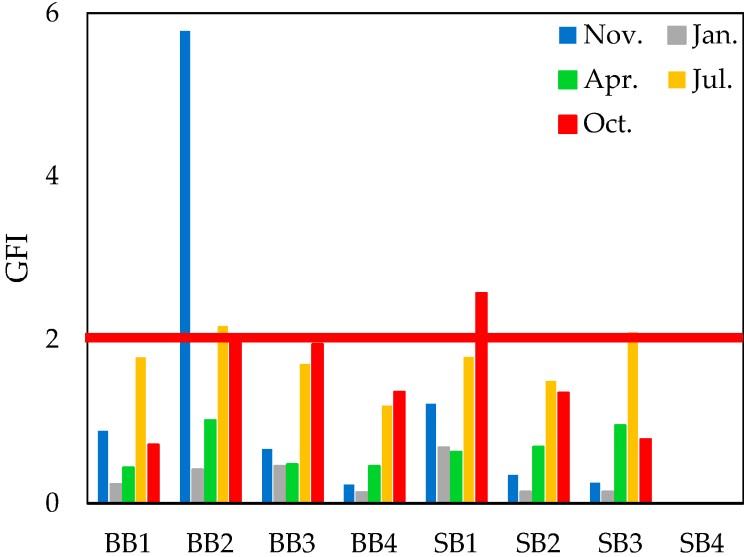

**Figure 4.** GFI for each sampled spring and sampling campaign (horizontal red line represents the value of GFI that indicates dominance of stygobite individuals).

*3.2. Biotic Variables*

3.2.1. Invertebrate Existence and Abundance

The results of the faunistic survey and monitoring of the eight springs distributed throughout BB and SB within five sampling campaigns showed that the sum of invertebrate individuals was 1390 with 750 stygobite and 640 non-stygobite individuals. In the following study, the invertebrates and stygobite individuals are presented in both time- and volume-integrated approaches. In general, the biotic parameters show significantly more spatial and temporal heterogeneity than the abiotic parameters.

With the time-integrated approach, the invertebrates can be visualized and evaluated in the study area and over the study period (existences). The existence of invertebrates was the highest in April and October with 83 Ind./d, and even in November (Figure 5 left), whereas the lowest was in July with 55 Ind./d. The existence of stygobite individuals was the highest in October with 60 Ind./d, whereas the lowest value was in the months of January, April, and July, with approx. 37 Ind./d (Figure 5 right). However, this information refers to all springs in the study area. The number of individuals per day at one spring was significantly lower (Figure 5). In the order BB2, SB2, SB1, BB1, BB3, BB4, SB3, and SB4, a decrease in the invertebrates and stygobite individuals was determined and expected according to the GFI. According to [48], the abundance of invertebrates and stygobite individuals (Ind/m$^3$) in our study was found to be uneven across the springs. It is notable for BB, however, that the springs on the eastern slope (BB1 and BB2) showed more individuals than on the western slope (BB3 and BB4). In the SB study area, SB1 in NW and SB2 in NE showed more individuals than SB3 in SE and SB4 in SW.

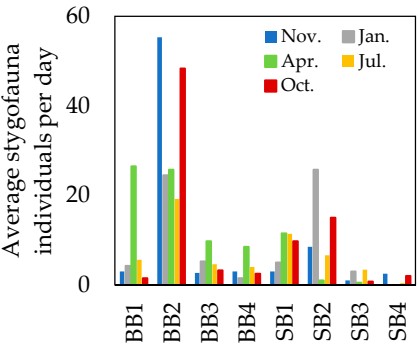 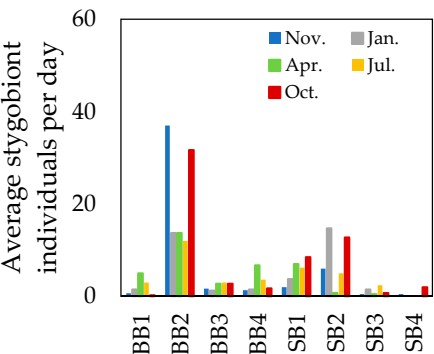

**Figure 5.** Average number of individuals of stygofauna (invertebrates) (**left**) and stygobites (**right**) per day for each sampled spring and sampling campaign.

Although invertebrates (stygofauna) were collected over 24 h, over 3–5 consecutive days, there were nonetheless 25 sampling events in which stygofauna and stygobite individuals were not found. In eleven sampling events, no evaluation of stygofauna sampling was possible because of a collapsed or detached net. In 25 sampling events, only one stygobite individual was counted. This shows the importance of establishing sampling times of 24 h on several consecutive days as a standard for stygofauna sampling at small natural springs.

With the volume-integrated approach, the abundance of invertebrates in the study area and over the study period can be visualized and evaluated. The abundance of invertebrates and stygobite individuals varied greatly between springs and sampling campaigns (Figure 6). The average invertebrate abundance was the highest in January with 1.18 Ind./m$^3$ and July with 1.12 Ind./m$^3$ (Figure 6 left), whereas the lowest was in April with 0.58 Ind./m$^3$. The abundance of stygobite individuals was the highest in July with 0.74 Ind./m$^3$, whereas the lowest value was in April with 0.25 Ind./m$^3$ (Figure 6 right). Thus, the abundance decreased with rising groundwater table and increasing spring discharge. In the order BB2, SB2, SB3, BB1, BB3, BB4, SB1, and SB4, a decrease in the abundance of stygofauna and stygobite individuals was determined.

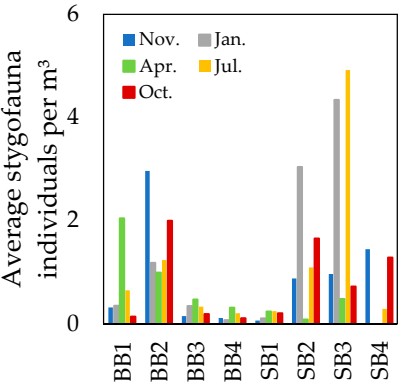 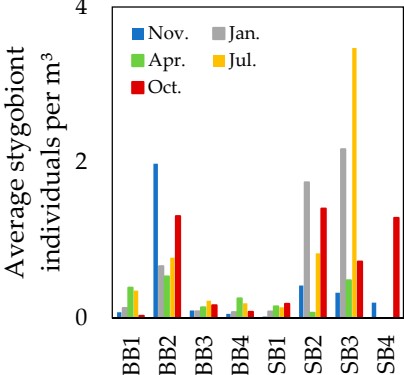

**Figure 6.** Average number of individuals of stygofauna (invertebrates) (**left**) and stygobites(**right**) per volume (Ind./m$^3$) for each sampled spring and sampling campaign.

The abundance of stygobite individuals, determined using the mentioned sampling technique, was mostly less than 2 Ind./m$^3$ and therefore less than 0.002 Ind./L. This means that the abundance was very much smaller than expected according to the GFI (Table 2). This discrepancy could be related to the anthropogenic effect of $NO_3$ available in the study area as mentioned before. This $NO_3$ is referred to in laboratory and field studies negatively affects stygobite availability in groundwater [15,56], even though recent investigations [57]

showed that this effect is less severe than expected. Thus, the discrepancy between GFI and the measured abundancy in the study area is most likely related to higher $NO_3$ concentrations which is not taken into account by GFI, and this index could be further checked and modified as also recommended in other studies [58].

3.2.2. Invertebrate Variability

A total of 750 stygobite individuals was recorded from eleven groundwater fauna groups (types or species) at eight springs in BB and SB during the five-day sampling campaign in five months. The non-stygobites (stygophile and stygoxene) were represented by fewer species, such as amphipods and isopods. Turbellarias and other insects, such as trichaoptera, are typically found in rheocrene springs [59]. For the following consideration of variability, all captured individuals were used for the evaluation.

According to the identification of DWA [50], the species of stygobite individuals were found for most of the meiofauna such as cyclopoids, harpacticoids, and ostracods, and macrofauna such as amphipods, isopods, and syncarida. Other species of macro and meio-fauna invertebrates, such as oligochaetes, diptera, turbellarian, gastropods, and nematodes, were also found in the studied springs. The types of stygobite individuals in the study area included cyclopoids and harpacticoids, while cyclopoids were the most dominant type in all studied springs, particularly BB2 and SB2, as cyclopoids are most tolerant against pollutants [60]. The harpacticoids and ostracods were also found, whereas other types were less common in other springs. It is already known that more chemically evolved groundwater suppresses stygofauna assemblages [15]. Electrical conductivity of groundwater was also reported to affect negatively the distribution of groundwater fauna [61]. The harpacticoids were not captured in most SB springs (Supplementary Figures S1 and S2 and Table S2), which might be related to the relatively higher salinity of SB in comparison to BB springs, as it was recently mentioned [62] that the harpacticoids are the most sensitive to salinity stress. Amphipods appeared only in BB4 and SB1, especially in July and October, respectively (Supplementary Figure S1 and Table S2). The variability of stygobite individuals, determined using the mentioned sampling technique, showed more than three taxa in most springs; this means that the variability is much higher than expected according to the GFI (Table 2).

It was also observed that the invertebrate types increased in July, then April and October, whereas they decreased in November and January (Supplementary Figure S2). The type of stygobite species, in contrast, increased in July and October and November, while it decreased in January and April The existence of more types seems to be related to a lower groundwater table and vice versa. This could be explained by the existence of a higher hydraulic gradient at the higher level of the groundwater table [63], which causes a higher groundwater flow velocity and intensive flushing of the available individuals. This in turn negatively influences their reproduction chance. The effect of the fluctuation in groundwater table on stygofauna assemblages was already referred to in other areas in the world [15], with no clear explanation. In Argentine, it was found that the thickness of the vadose zone affects inversely the abundance of invertebrates in groundwater, because a thicker vadose zone hinders sufficient amount of organic matter from reaching the groundwater [64]. Experimental studies also showed that the decrease in sediment water content due to the decline of water table in general increased the mortality rate of stygofauna, with a different response of some taxa [41]. However, all these investigations do not explain the appearance of more species, reported in the research, at the lower stand of groundwater table.

The data of invertebrates and stygobite individuals showed temporal variances in species abundance. This observation is related to numerous environmental factors that can be addressed by statistical analysis. Spring BB2 usually had many animals of both types, stygobites and non-stygobites, whereas SB4 had a smaller number, and in some cases these species might be absent. The reason for this might be the urban influence (anthropogenic

effect) and the lake of detritus. In contrast, other studies of spring water located in nearby rural areas identified typical numbers of stygofauna individuals [54].

*3.3. Statistical Analysis*

3.3.1. Spearman Correlation

Spearman correlation ($n = 80$, $p \leq 0.01$) was conducted between different biotic and abiotic data collected during the sampling campaigns, and the results for the sampling springs of Baumberge and Schöppinger Berg are shown in Supplementary Table S3 and Table S4, respectively.

The results of the correlation analysis of the BB springs data show very significant positive correlations between all of the biotic parameters, such as quantity of invertebrates (stygofauna) and stygobite individuals overall, per day ($\overline{I_{sf\_d}}$ and $\overline{I_{sb\_d}}$), and per cubic meter ($\overline{I_{sf\_V}}$ and $\overline{I_{sb\_V}}$). An increase in the quantity of invertebrates and stygobite individuals in groundwater causes more individuals to be flushed out with discharging groundwater through the springs. This interrelationship can also be proven by the positive correlation between the amount of stygobite individuals per day ($\overline{I_{sb\_d}}$) and the level of the groundwater table. All of the mentioned biotic parameters correlate positively with detritus as the main source of organic matter, such as food, in BB. It is remarkable that only the quantity of stygobite individuals per cubic meter shows a significant positive correlation with GFI; thus, GFI provides no clear indication of the potential for the presence of stygofauna in BB. As expected, GFI correlates positively with detritus (Equation (3)), but also negatively with EC and $h_{GW}$. Thus, a decrease in electrical conductivity, e.g., as a result of rainwater dilution, means an increase in the detritus of discharging spring water. The negative correlation between GFI and $h_{GW}$ could be an indicator that an increase in hydraulic head in the aquifer is a complex process and rainfall percolating downward is not the main cause of this increase. Alternatively, this could also mean that more stygofauna individuals can be expected if the groundwater level is higher and/or conductivity is lower.

Regarding the other abiotic parameters, a significant positive correlation exists between $h_{GW}$ and $V$ and EC; rainwater percolating from the surface dissolves the available constituents in the vadose zone and the EC of groundwater increases accordingly, and the groundwater table raises and also causes an increase in the hydraulic gradient and in the spring discharge.

The results of the correlation analysis of the SB spring data show a less significant positive correlation between the quantity of stygofauna and stygobite individuals overall and the quantity of stygofauna and stygobite individuals per day ($\overline{I_{sf\_d}}$ and $\overline{I_{sb\_d}}$). All of these biotic parameters show a significant positive correlation with DO because DO is one of the controlling factors for the existence of stygofauna and stygobite individuals. Only the quantity of stygobite individuals per day ($\overline{I_{sb\_d}}$) correlates positively with detritus.

The abundances of stygofauna and stygobite individuals ($\overline{I_{sf\_V}}$ and $\overline{I_{sb\_V}}$) show a very significant positive correlation with each other, but not on any other parameter. Regarding the abiotic parameters, as expected, GFI correlates positively with detritus (Equation (3)) but, as in BB, also negatively with EC. The negative correlation between detritus and electrical conductivity also confirms in SB that a decrease in electrical conductivity, e.g., as a result of rainwater dilution, means an increase in the detritus of discharging spring water. In SB, detritus correlates positively with DO and $V$; detritus and DO are transported from the surface with the water percolating downward, and an increase in the discharge means less chance for DO to be consumed in the aquifer.

In summary, comparable dependencies can be identified in the two study areas, but their parameters differ in relevance. In BB, detritus is an important factor, whereas DO is an important factor in SB.

3.3.2. Factor Analyses

From the abiotic and biotic parameters (14 variables) of 80 spring samples, three and four factors (PCs) were extracted, explaining about 74% and 71% of total sample

variance (Tables 5 and 6) for BB and SB, respectively. The downward curve scree plots and component plots in rotated space of principal component loadings for both groups of springs are shown in Supplementary Figure S3 and Figure S4. For BB, the percentages of variance explained by the factors are 41.8% for PC1, 12.4% for PC2, 10.7% for PC3, and 9.4% for PC4, whereas for SB, the percentages of variance explained by the factors are 29.8% for PC1, 16.1% for PC2, 15% for PC3, and 9.9% for PC4. High positive loadings (Pearson correlation coefficient) of different variables on the same factor may indicate a close relationship among the respective variables, whereas negative loadings between variables indicate an inverse relationship.

**Table 5.** Principal component loadings from 14 variables for 80 samples collected from November 2018 to October 2019 in five sampling campaigns in Baumberge (correlation coefficients of loadings at corresponding factors are significant at $p \leq 0.01$ (two-tailed)).

| Parameter | Component | | | |
|---|---|---|---|---|
| | PC1 | PC2 | PC3 | PC4 |
| $sf_{TOT}$ | 0.95 | | | |
| $\overline{I_{sf\_V}}$ | 0.93 | | | |
| $sb_{TOT}$ | 0.92 | | | |
| $\overline{I_{sb\_V}}$ | 0.92 | | | |
| $\overline{I_{sf\_d}}$ | 0.87 | | | |
| $\overline{I_{sb\_d}}$ | 0.85 | | | |
| Detritus | 0.72 | | | |
| GFI | | 0.52 | | |
| pH | | 0.71 | | |
| Temp. | | 0.68 | | |
| $V$ | | | 0.8 | |
| $h_{GW}$ | | | 0.73 | |
| DO | | | | 0.79 |
| EC | | | | 0.72 |
| **% Variance** | 41.8 | 12.4 | 10.7 | 9.4 |
| **% Cumulative** | 41.8 | 54.2 | 64.9 | 74.3 |

**Table 6.** Principal component loadings from 14 variables for 80 samples collected from November 2018 to October 2019 in five sampling campaign in Schöppinger Berg (correlation coefficients of loadings at corresponding factors are significant at $p \leq 0.01$ (two-tailed)).

| Parameter | Component | | | |
|---|---|---|---|---|
| | PC1 | PC2 | PC3 | PC4 |
| $\overline{I_{sf\_d}}$ | 0.92 | | | |
| $\overline{I_{sb\_d}}$ | 0.91 | | | |
| $sf_{TOT}$ | 0.87 | | | |
| $sb_{TOT}$ | 0.85 | | | |
| DO | 0.73 | | | |
| $\overline{I_{sb\_V}}$ | | 0.94 | | |
| $\overline{I_{sf\_V}}$ | | 0.94 | | |
| GFI | | | 0.88 | |
| Detritus | | | 0.82 | |
| $V$ | | | 0.63 | |
| pH | | | | 0.73 |
| EC | | | | 0.71 |
| Temp. | | | | 0.5 |
| $h_{GW}$ | | | | |
| **% Variance** | 29.8 | 16.1 | 15 | 9.9 |
| **% Cumulative** | 29.8 | 45.9 | 60.9 | 70.8 |

Table 5 shows PC results of BB, where four PC are extracted that explain 74.3% of the total variance. The quantity of stygofauna and stygobite individuals overall, per day ($\overline{I_{sf\_d}}$ and $\overline{I_{sb\_d}}$), and per cubic meter ($\overline{I_{sf\_V}}$ and $\overline{I_{sb\_V}}$), in addition to detritus, have positive and significant ($p \leq 0.01$) loadings on PC1. This factor is in accordance with the results of the Spearman correlation. Detritus plays a key role in the abundance of stygofauna and stygobite individuals, as shown in previous research [51].

GFI, pH, and Temp. show significant positive loadings on PC2. This means that GFI, pH, and Temp. increase and decrease simultaneously as they are affected by the same controlling factors (most likely water percolating downward into the aquifer and the residence time within the aquifer, which is inversely correlated with the spring discharge). Excluding any atmospheric effect during sampling and on-site measurement, this factor is indicative of the interaction between surface- and groundwater. The third factor (PC3) shows significant positive loadings of $V$ and $h_{GW}$, which is explained by the increase in the hydraulic gradient in the aquifer with the increase in the hydraulic head and, consequently, groundwater flow velocity and $V$. The fourth factor (PC4) shows significant positive loadings of DO and EC, which is highly indicative of the key role of the interaction between rainwater infiltration and groundwater (excluding any atmospheric effect during sampling and measuring of on-site parameters).

In the case of SB, four PCs were extracted which explain 70.8% of the total variance (Table 6). The quantity of stygofauna and stygobite individuals overall, per day ($\overline{I_{sf\_d}}$ and $\overline{I_{sb\_d}}$), and DO load are significant ($p \leq 0.01$) on PC1. This factor once again highlights the role of DO on the availability of stygofauna and stygobite individuals in groundwater in this area [64]. Surprisingly, stygofauna and stygobite individuals per cubic meter ($\overline{I_{sf\_V}}$ and $\overline{I_{sb\_V}}$) load solely on PC2, and not on PC1, which means that discharged stygofauna and stygobite individuals are not controlled by their existence in groundwater, but by the amount of discharged groundwater. Detritus, GFI, and $V$ show significant positive loadings on PC3, which can be explained by the role of discharge in enhancing the spatial distribution of detritus in the aquifer [65]. The fourth factor (PC4) shows significant positive loadings of pH, EC, and Temp. Actually, groundwater in the study area is of Ca-HCO$_3$ type as a result of carbonate dissolution, which is also supported by a significant positive correlation between EC and Ca. Excluding any atmospheric effect during sampling and measuring of on-site parameters, the fourth factor is highly indicative of the key role of the interaction between rainwater/groundwater and the sediments that make up the aquifer [63].

In general, factor analysis provides some hints but no clear indications of the processes that influence the existence and abundance of sampled stygofauna and stygobite individuals; for example, the role played by the internal morphology of the aquifer, and its trapping function of stygofauna and stygobite individuals in some dead pores or small fissures, cannot be captured statistically [66].

## 4. Conclusions

The abiotic and biotic conditions of spring water in eight springs in Baumberge and Schöppinger Berg (Münsterland area, North-Rhine Westphalia in Germany) were investigated in this research, via sampling five times between 18 November and 19 October Some springs were found to be devoid of invertebrates (SB4), whereas others showed a high count of invertebrates and stygobites (BB2 and SB2). More invertebrates were found in some springs (BB4) and months (19 July). In small natural springs it is important to establish sampling periods of 24 h over several consecutive days as a standard for invertebrate sampling.

In general, the abundance of invertebrates increased for higher groundwater tables and increasing spring discharge, and decreased for lower groundwater tables. In BB, the presence of stygobite individuals per day increased for higher groundwater tables because high flow velocities induced by high hydraulic head and spring discharge most likely help in flushing stygofauna from more distant areas and stagnant zones. In contrast, higher

number of species was evident at lower groundwater tables. The high flow velocities most likely result in some groups being captured in some dead zones, e.g., smaller fissures and dead-end pores. Therefore, the presence of stygofauna appears to be greater in lower groundwater tables and lower groundwater flow velocities. Thus, the optimal time to study biodiversity is when the groundwater table is lower, whereas the presence of stygobite individuals is better assessed when the groundwater table is higher. Therefore, investigating the abundance of invertebrates and invertebrate species is not recommended to be done collectively but separately.

In the light of measured on-site parameters, it is difficult to draw a conclusion about the actual role of invertebrates as environmental indicators; however, comparing the GFI with the measured availability of invertebrates points to a possible factor—$NO_3$ input from the application of fertilizers—which alters negatively the existence of invertebrates, shifting their distribution patterns and possibly eliminating some taxa.

The Spearman correlation test (two-tailed) and factor analyses ($n = 80$, $p \leq 0.01$) of BB highlighted the importance of detritus as the main controlling factor of the occurrence of stygofauna and stygobite individuals, whereas dissolved oxygen was found to be essential for the occurrence in SB. The abundance of stygofauna and stygobite individuals is affected by different factors in the two areas.

**Supplementary Materials:** The following are available online at https://www.mdpi.com/2073 -4441/13/3/359/s1, Table S1. Geographical coordinates, elevation, and catchment of the eight selected sampling springs in the two study areas of Baumberge and Schöppinger Berg, Table S2. Abundant taxa recorded in Baumberge and Schöppinger Berg springs within five sampling campaigns (November 2018, January, April, July and October 2019), Table S3. Spearman correlation coefficient matrix (correlation coefficients are significant at $p \leq 0.01$ (two-tailed)) for biotic and abiotic parameters in the BB springs at Baumberge from November 2018 to October 2019 in five sampling campaigns, Table S4. Spearman correlation coefficients matrix (correlation coefficients are significant at $p \leq 0.01$ (2-tailed)) for biotic and abiotic parameters in the SB springs at Schöppinger Berg from November 2018 to October 2019 in five sampling campaigns, Figure S1. Matrix of bar plots of different types of invertebrates for each sampled spring, Figure S2. Matrix of bar plots of different types of invertebrates for different sampling campaigns, Figure S3. Scree plot (right) and component plot in rotated space (left) of principal component loadings of 14 variables for 80 samples collected from November 2018 to October 2019 in five sampling campaigns in Baumberge (horizontal red line: eigenvalue = 1), Figure S4. Scree plot (right) and component plot in rotated space (left) of principal component loadings of 14 variables for 80 samples collected from November 2018 to October 2019 in five sampling campaigns in Schöppinger Berg (horizontal red line: eigenvalue = 1).

**Author Contributions:** Conceptualization, S.A.A. and P.G.; Field work, S.A.A.; Data curation, S.A.A. and P.G.; Funding acquisition, S.A.A.; Methodology, S.A.A., W.K., and P.G.; Resources, S.A.A., and P.G.; Supervision, P.G.; Writing—original draft, S.A.A., W.K., and P.G.; Writing—review and editing, S.A.A., W.K., and P.G.; All authors have read and agreed to the published version of the manuscript.

**Funding:** This work was funded by the Hans Böckler Foundation.

**Acknowledgments:** We are grateful for financial support through individual doctoral funding from Hans Böckler Foundation. Thanks to Spengler for the help during the Groundwater ecology workshop at the Koblenz-Landau University. The authors are indebted to Hans Jürgen Hahn for the inspiring, generous, and stimulating discussions. Our thanks go to the anonymous reviewers who helped to improve this article.

**Conflicts of Interest:** The authors declare no conflict of interest. The funder had no role in the design of the study; in the collection, analyses, or interpretation of data; in the writing of the manuscript, or in the decision to publish the results.

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
