# Peer review of "Comparative Investigation of Aquatic Invertebrates in Springs in Münsterland Area (Western Germany)"

_water, doi:10.3390/w13030359_

Round 1

Reviewer 1 Report

In my opinion the paper presents an interesting research topic referring to the occurrence of invertebrates in spring water.  Stressing their specific character and place in the ambient environment seems important as shown by relevant data provided in the manuscript. The organization and the style of the paper is appropriate, however , some minor revisions are needed before the paper is processed further.

General comments

The introduction should contain more definite information on the role and the significance of invertebrates, especially as environment quality indicators ( 3- 4 sentences). Some brief reference to the „state of the art” found after literature search should also be provided, as it seems a little missing currently. If possible, some information/ relevance to other studies in order to make it more global problem- not only restricted to two mountainous research sites.

The number of data used in statistical analyses is sometimes missing ( detailed comments). It would be also better, if some correlation diagrams ( graphs) are provided ( at least for 2-3 variables) for visual estimates of the dependence between them.

It is a little questionable (in the abstract and the data analysis) to be sure about the influence of groundwater head on invertebrates quantity and so on. I may be wrong, but better say that most likely or probably there was the impact, since you have insufficient groundwater table data. It would be good to have the general map of groundwater izopotentials ( equal heads) to say little about the directions of groundwater flow and the water table gradient, rather than base only on a single well.

Specific comments:

Line 212- Figure 2 can be skipped. Basing only on a few values does not deserve a picture. It is recommended to leave only the description in lines 215-219. Figure 3 is ok., it shows a number of values and makes clear view of the spring discharge varibility.

Line 232- I would rather write ,, temporal” fluctuation. You have two reserch spots , in fact, nevertheless spatial changes are to be estimated over a number of points – like interpolated on a map over a wider area or so.

 Line 243 – you’d better skip figure 4 – the data can be given in a table.

Line 292 - 301– Why do you use the term abundance? Does it mean too many, or too much? Or excess? A reference is needed or criteria to state which quantity is normal , and which is not.

Lines 317- 333 – Are there some literature references or other case studies available to determine if it is really an expected situation? And is the groundwater quality important here?

Line 350 – How many measurements of grondwater table were used? You only provided 4 values in the paper ( Table 2 and 4) .

Gerenal remars until line 370- Please provide the number of values used for the correlation analysis. Unless it is unknown, the sense of the analyses seems mysterious.

Lines 441-443 How do you determine and define the residence time and its’ relationship to spring discharge? Is there any influence of underlying rocks ( minerals) on water quality – e.g. EC and pH ? – Supposing the horizontal movement of water domiantes over a vertical one- which is only an assumption- the quality of water would be the result of the contact with the minerals.

Author Response

Wo would like to thank you very much for dedicating part of your valuable time to review our manuscript, and helping us through your experience to improve our manuscript qualititavely. Attached is a word file listing your valuable comments and hints and our answers to them marked in yellow colour.

Reviewer 2 Report

Manuscript number 1029146

Title: Comparative investigation of invertebrate in spring water in Baumberge and Schöppinger Berg mountains, Münsterland area in western Germany

The authors studied invertebrate communities in groundwater on high mountain springs along environmental gradient. The proposed topic is interesting, studies on stygofauna assemblages are still rare and insufficient.

I have some suggestions for the manuscript before it will be accepted for publication.

Introduction

Some recent publication on relationships between environmental factors and stygofauna communities should be added to better support the purpose of the study.

The purpose of the study should be precisely formulated at the end of Introduction.

Materials and methods

Field work

Page 4, lines 111-112

Sampling were carried out from November 2018 to October 2019. There is no need to use Nov 18, Jan 19, etc. In text and figures, please use Nov instead of Nov 18, Jan instead of Jan 19. Years are not necessary, only months are important.

Please explain differences in sampling intervals, two or three months

Page 4, Lines 119-120

Samples were colleted using plankton net of 74 µm mesh size. Whether this mesh size is suitable for collecting invertebrate fauna of such a wide range of body ? Please explain.

How the numer of individuals was estimated? On Figures it is numer of individuals per day, what was the reference unit, cm3, ml?

Figures

Please delete „Sampling campain” from Figures 2, 3 and 4. Names of months on X axis are sufficient.

Please add values of standard error (±SE) or standard deviation (±SD)on Figures 2, 3 and 4

Results and Discussion

Invertebrate variety

please change to Invertebrate variety to Variability of invertebrates

Page 15, lines 317-320

Authors stated that „Other species such as oligochaetes and hirudinea from the phyla annelida (worms), mollusca (snails and slugs) taubullaren and nematodes (roundworms) were also found in the studied springs.  How were these groups of organisms identified?

Page 15, Lines 334-335

„…The data of invertebrates and stygobites individuals showed temporal variances in abundances of species….”

To what systematic order were particular groups of organisms determined, taxon or species? Please be specific. It is not possible to compare the diversity of organisms when they are identified to different systematic orders

Please formulate at the end of Results and Discussion, a separate paragraph with concluding remarks, to sum up the study.

The manuscript should be verified by native English spea

Manuscript number 1029146

Title: Comparative investigation of invertebrate in spring water in Baumberge and Schöppinger Berg mountains, Münsterland area in western Germany

The authors studied invertebrate communities in groundwater on high mountain springs along environmental gradient. The proposed topic is interesting, studies on stygofauna assemblages are still rare and insufficient.

I have some suggestions for the manuscript before it will be accepted for publication.

Introduction

Some recent publication on relationships between environmental factors and stygofauna communities should be added to better support the purpose of the study.

The purpose of the study should be precisely formulated at the end of Introduction.

Materials and methods

Field work

Page 4, lines 111-112

Sampling were carried out from November 2018 to October 2019. There is no need to use Nov 18, Jan 19, etc. In text and figures, please use Nov instead of Nov 18, Jan instead of Jan 19. Years are not necessary, only months are important.

Please explain differences in sampling intervals, two or three months

Page 4, Lines 119-120

Samples were colleted using plankton net of 74 µm mesh size. Whether this mesh size is suitable for collecting invertebrate fauna of such a wide range of body ? Please explain.

How the numer of individuals was estimated? On Figures it is numer of individuals per day, what was the reference unit, cm3, ml?

Figures

Please delete „Sampling campain” from Figures 2, 3 and 4. Names of months on X axis are sufficient.

Please add values of standard error (±SE) or standard deviation (±SD)on Figures 2, 3 and 4

Results and Discussion

Invertebrate variety

please change to Invertebrate variety to Variability of invertebrates

Page 15, lines 317-320

Authors stated that „Other species such as oligochaetes and hirudinea from the phyla annelida (worms), mollusca (snails and slugs) taubullaren and nematodes (roundworms) were also found in the studied springs.  How were these groups of organisms identified?

Page 15, Lines 334-335

„…The data of invertebrates and stygobites individuals showed temporal variances in abundances of species….”

To what systematic order were particular groups of organisms determined, taxon or species? Please be specific. It is not possible to compare the diversity of organisms when they are identified to different systematic orders

Please formulate at the end of Results and Discussion, a separate paragraph with concluding remarks, to sum up the study.

The manuscript should be verified by native English speaker.

Author Response

(The authors gave the same response as above.)

Reviewer 3 Report

The manuscript presents a study of aquatic invertebrates in springs, which are strategic as sources of potable water and vulnerable habitats at the same time. We can hardly think of a habitat that we wish more to be in optimal conditions as springs. So, the topic is of a great interest. Authors performed numerous statistical analyses on samples, but on the other hand determined the sampled invertebrates only to higher taxonomic units. I suggest the authors to get more information on the taxonomic composition, since the total number of individuals is much lower than in superficial habitats. According to this and concerns described below I suggest to greatly elaborate the manuscript and submit it again.

Authors are suggested to send their manuscript to english-editor to improve the language.

The present title contains four geographical terms which distract the potential readers from the contribution.  

»Comparative investigation of invertebrate in spring water in Baumberge and Schöppinger Berg mountains, Münsterland area in western Germany.« I suggest shortened title:  »Comparative investigation of aquatic invertebrates in springs in Münsterland area (western Germany)

The text should be better balanced - consistent introduction with sufficient literature review is followed by extensive chapter Materials and methods and chapter Results and discussion, which extend on 12 pages. I suggest to shorten the manuscript for 25%. I also suggest to divide the Results and Discussion in two separate chapters.

Pagination of the manuscript should be solved as well.

Authors cite 39 references, but only four of them are used in Discussion, two of them being from the same researcher (which the corresponding author knows personally) and only two of them referring to invertebrates, which are the focus of the manuscript. This is not enough. The results are actually not discussed in the context of previous researches. I advise the authors to compare their results with results of contributions used in Introduction and to search for additional suitable literature to put their results into wider context. For instance, authors do not compare and discuss the methods of other researchers while their key outcome is the most suitable time of sampling, etc.

The names of invertebrate taxa are frequently misspelled or written completely wrong. E.g. figures 8 and 9: Amphiopds – Amphipods, Syncardia – Syncarida, Tubellaria – Turbellaria.

Ln 319: taubullaren and nematods (roundworms) – turbellarians (flatworms) and nematods (roundworms), (names of higher taxa in brackets are not necessary),

This makes also the determination of the sampled invertebrates questionable. I suggest the authors to consider this shortcoming and ask for help of a colleague which works in the research field of invertebrate ecology. One solution could be the inclusion of additional author from the field of Biology or Ecology, maybe one of those written in Acknowledgement, which would take care of this issue.

The authors conclude that the samplings of invertebrate abundance and invertebrate species should be done separately. When we study the communities of aquatic invertebrates we are usually interested in both characteristics, but they are sampled at the same time. If the reason for sampling these two characteristics separately the authors should define what could such monitoring enable. Aquatic invertebrates are often used as bioindicators of the conditions in aquatic ecosystems and enable the assessment of ecological status. I invite the authors to explain these possibilities.

Mscr. must contain the paragraph Author contributions, where they define the tasks performed by each of the coauthors. Use also “We are” grateful in Acknowledgments.

Minor comments:

Tables 6 and 7 should be prepared in accordance with instructions (without vertical lines). I recommend to place them in Appendix.

Authors should cite the tables and figures in the text in the way described in instructions.

In the Figure 1 you use degrees, minutes and seconds, while in the text Lines 105-106 and in Table 1 the coordinates are written in different form. Please use the abbreviations N and E for latitude and longitude (51° 58’ N; 7° 21’ 30’’ E).

There is no need to write brackets used for units after pH (e.g. line 127, Table 5,)

Ln 248: pH was almost circumneutral (pH=7)... --> write: mean values of pH were 7.2 ….

Ln 252: Figure 5. Explain the horizontal red line in caption.

Ln 305: Invertebrates variety --> Diversity of invertebrates

Ln 478: high count of invertebrates and stygobites – it is written like that stygobites are not invertebrates

Ln 486: higher abundance if species --> higher number of species

References:

Please check the references thoroughly. There are too much mistakes to list them.

Author Response

(The authors gave the same response as above.)

Round 2

Reviewer 1 Report

Lines 442- 444 Please clarify if this is only the role of interaction between rain water and groundwater. This refers to the last point of the coverletter ( point no. 11) regarding the impact of existing rock deposits on groundwater chemistry.

Please kindly provide the number of grounwater table observations in the section 3.3.2 ,,Factor analyes" The number of samples is stated ( 80 spring water samples) but the information only about four groundwater table measuremets remains.

There should be one general conclusion added, concerning the meaning/role of invertebrates as environment quality indicators against available literature knowledge- was it sufficient? limitations? - a comment at the background of the literature search. The conclusions on the achieved results are sufficient and comprehensible, but a further context on the base of the existing knowledge would be recommended.

Author Response

Please accept our thanks for dedicating part of your valuable time to process our manuscript, and helping us through your experience to improve our manuscript qualitatively. Attached is a coverletter listing all your comments and remarks (in black colour) and our answers to them (in red colour). Our changes in the main manuscript are highlighted in yellow colour.

Reviewer 2 Report

Manuscript number 1029146

Final comments

Title:

Comparative investigation of invertebrate in spring water in Baumberge and Schöppinger Berg mountains, Münsterland area in western Germany

The Authors revised the manuscript referring to the review comments.

The manuscript can be published in present form.

Author Response

Please accept our thanks for dedicating part of your valuable time to process our manuscript, and helping us through your experience to improve our manuscript qualitatively. 

Reviewer 3 Report

The manuscript has been improved. I suggest the authors to continue with the improvement of their manuscript, according to the concerns and comments below.

Concerns:

Authors performed numerous statistical analyses on samples, but on the other hand determined the invertebrates only to higher taxonomic units. I suggest the authors to present more information on the taxonomic composition in a condensed way - in a table with information about the invertebrate community for both study areas and for 5 sampling campaigns.  

The results of this study are only poorly discussed in the context of previous researches.  Although the authors have added 21 more references there is still a lack of their use and integration in Discussion, since only 5 more are efficiently used to discuss the results of their study. More than half of the paragraphs in R&D chapter do not contain any citation so the results are not in the broader scientific context. I suggest the authors to use the collected literature which is cited mostly in introduction, to read their findings and conclusions and compare them with their own results. Only five of them referring to invertebrates, which are the focus of the manuscript, which is still not enough.

Minor comments:

Ln 274: Figure 4. red line in caption do you mean: “red line represents the value of GFI that indicates dominance of stygobite individuals”.  Please rewrite.

References:

Check the references again. There are still many mistakes there, e.g. In some references first names are not abbreviated and not behind the family name; there are some words inserted that do not belong there etc.

Author Response

(The authors gave the same response as above.)

Round 3

Reviewer 3 Report

The Results and discussion section has been improved. I suggest the authors to perform the minor improvements of their manuscript, according to the comments below.

comments:

I suggest the authors to present more information on the taxonomic composition in a condensed way - in a table with information about the invertebrate community for both study areas and for 5 sampling campaigns.  

References:

There are still some mistakes, e.g.

ref 8: delete chapter 11

ref 9: insert editors

ref 17: abbreviate first names

ref 18: correct ISBN

ref 24: correct the title

Author Response

Please accept our thanks for dedicating part of your valuable time to process our manuscript, and helping us through your experience to improve our manuscript qualitatively. Attached is a coverletter listing all your comments and remarks (in black colour) and our answers to them (in red colour). Any changes in the manuscript and in the supplementary file are highlighted in yelow colour.
